# Drivers of 'voluntary' recruitment and challenges for families with adolescents engaged with armed groups: Qualitative insights from Central African Republic and Democratic Republic of the Congo

**Alexandra H. Blackwell**[1,2]*, **Yvonne Agengo**[3], **Daniel Ozoukou**[1,4], **Julia Ulrike Wendt**[5], **Alice Nigane**[6], **Paradis Goana**[6], **Bertin Kanani**[5], **Kathryn Falb**[1]

1 Airbel Impact Lab, International Rescue Committee, Washington, DC, United States of America, 2 Department of Social Policy & Intervention, University of Oxford, Oxford, United Kingdom, 3 Violence Prevention & Response Unit, International Rescue Committee, Geneva, Switzerland, 4 Zolberg-IRC Fellow, The New School, New York City, New York, United States of America, 5 Violence Prevention & Response Unit, International Rescue Committee, Goma, North Kivu, Democratic Republic of Congo, 6 Violence Prevention & Response Unit, International Rescue Committee, Bangui, Central African Republic

* alexandra.blackwell@spi.ox.ac.uk

## Abstract

Globally, armed conflicts have increased threefold since 2010. The number of children voluntarily engaging with armed groups is also rising, despite increasing efforts to prevent this grave human rights violation. However, traditional approaches focusing on the prevention, release, and reintegration of children through forced recruitment do not adequately address the complex and interlinking push and pull factors of voluntary recruitment. This qualitative study sought to deepen understanding of the drivers and consequences of voluntary recruitment from the perspectives of adolescents and their caregivers, as well as to explore how to better support families living in conflict settings. In-depth interviews were conducted with 74 adolescents (44 boys and 30 girls) ages 14 to 20 years and 39 caregivers (18 men and 21 women) ages 32 to 66 years in two distinct conflict settings: North Kivu, Democratic Republic of Congo and Ouham-Pendé, Central African Republic. Interviews with adolescents utilized a visual narrative technique. The findings examine the unique perspectives of adolescents engaged with armed groups and their caregivers to understand how conflict experiences, economic insecurity, and social insecurity influence adolescent's engagement with armed groups and reintegration with their families. The study found that families living in conflict settings are subject to traumatic experiences and economic hardship that erode protective family relationships, leaving adolescent boys and girls particularly vulnerable to the systemic and overlapping factors that influence them to engage with and return to armed groups. The findings illustrate how these factors can disrupt protective social structures, and inversely how familial support can act as a potential protective factor against recruitment and break the cycle of reengagement. By better understanding the experiences of adolescents enduring recruitment and how to support caregivers of those adolescents, more comprehensive programming models can be developed to adequately prevent voluntary

**Data Availability Statement:** The data generated and/or analysed during the current study are not publicly available due to the sensitivity of the data but are available from the corresponding author or the International Rescue Committee Institutional Review Board on reasonable request via email (humansubjects@rescue.org).

**Funding:** This research was made possible through a grant by the United Stated Agency for International Development (720FDA19GR00218). The contents are the responsibility of the International Rescue Committee and do not necessarily reflect the views of USAID or the United States Government. AB, YA, JUW, AN, PG, BK, and KF received partial funding for their salaries from the USAID as part of this grant. DO was funded by the IRC-Zolberg Fellowship.

**Competing interests:** The authors have declared that no competing interests exist.

recruitment and promote successful reintegration, enabling children to reach their full potential.

## Introduction

The number of children living in conflict zones has increased almost threefold, from under 5% of children in 1990 to more than 14% in 2019 [1]. In 2020, one in five children living in a conflict zone globally was living in an area with reports of child recruitment by conflict actors [2]. This persistent and growing problem has been documented in conflicts around the world including the Democratic Republic of Congo (DRC), Central African Republic (CAR), Iraq, Syria, and numerous other conflict-affected regions. Formerly termed 'child soldiers', 170 countries have ratified an international agreement to forbid the recruitment and use of children in any kind of regular or irregular armed force or armed group in any capacity, including the prohibition of anyone under the age of 18 being conscripted into state forces or engaging with any non-state armed group [3]. While some progress has been made, there is a growing number of children being driven to engage with armed forces and armed groups [4].

### 'Voluntary' recruitment of children to armed groups

While forced child recruitment through abduction and threats of violence still persists, in an increasing number of conflict settings, boys and girls exercise some level of agency in their decision to join armed groups, even while living with their families [4]. For the purposes of this study, 'voluntary' recruitment refers to situations where children are not abducted or taken away by armed actors; however, the authors acknowledge that children do not truly 'volunteer' because either they do not have a choice (for example, they are pushed by their economic situation) or they do not fully understand the implications of being part of an armed group given their developmental stage. Adolescents aged 13–18 years are particularly vulnerable to voluntary recruitment due to their stage of cognitive development and transitional role in society [5], in which adolescents are still in the process of acquiring the ability to make rational decisions and have not yet fully developed the ability to understand mortality, leading to easier 'voluntary' recruitment and manipulation to engage in violence or extremist indoctrination. Within child development theory, adolescence is the transitional psychosocial stage between childhood and adulthood [6] which can occur from around ages 10 to 24 [7]. For the purpose of this study, we use the term "adolescents" to refer to a specific subgroup of older children and young people within this transitional period, whereas the term "children" is used more broadly to refer to any age group of children from birth up to 18 years. While in many contexts including the DRC and CAR, adolescents are considered adults within society based on cultural rites of passage and initiation, international human rights and humanitarian law prohibits any recruitment or use of persons under the age of 18 by armed groups and bans the forced recruitment and participation of children under 18 in state militaries, and no distinction is made between 'voluntary' and forced recruitment of this age group [8].

Beyond forced recruitment and manipulation into armed groups, three other pathways have been documented for 'voluntary' drivers of engagement [9]. First, there are powerful economic drivers through which children may join armed groups to improve their economic wellbeing through potential monetary incentives [10]. This may be a particularly influential driver in settings with chronic poverty, disruptions to schooling, and broader insecurity that are often typified in conflict-affected settings. A key second driver involves an ideological or avenging component to joining armed forces [10]. This is an increasingly common pathway to

engagement with armed forces, as a growing contingent of youth are increasingly engaged in radicalization and violent extremism [11]. For example, in northern Uganda's civil conflict, rebel leaders exploited the ability to manipulate children through misinformation and indoctrination to recruit and retain children. Finally, in some settings there may also be an instrumental component to joining armed groups around the 'guarantee' or provision of safety or security provided by alignment with armed groups [12,13]. Ultimately, literature demonstrates that multiple factors influence the likelihood that a child would join an extremist or armed group or other armed forces, including upbringing, education, governance, and sense of belonging [14], and therefore more evidence is needed to explore how these push and pull factors overlap and how potential protective factors can support a child to decide not to engage with an armed group.

## Socioecological theories on the experience of children and families in conflict

Existing socioecological theories [15–17] on the effect of conflict on child mental health and psychosocial wellbeing depict how risk and protective factors at different socioecological levels interact to drive recruitment into armed groups. These models emphasize the importance of child development and how risk factors introduced by conflict (e.g., traumatic war-related experiences, displacement, abduction, and community violence or affiliations, etc.) can directly result in poor developmental outcomes. Conflict can also indirectly exacerbate non-conflict risk factors (e.g. poverty and socioeconomic status, breakdown in social, economic, and service structures, and suppressed civil society, etc.) which in turn negatively affect risk factors at the interpersonal and individual levels. In contrast, protective factors at different socioecological levels could contribute to the strength and coping of young people living in conflict [15]. While empirical evidence examines some of these push and pull factors of child recruitment at the macro- and societal levels, there are few studies examining risk and protective factors influencing voluntary child recruitment at the household and interpersonal level, and how these factors vary throughout the cycle of a conflict. Recent evidence has identified the parent-child relationship as a potential deterrent and protective factor for children. For example, one recent study underscored the perceived lack of parental involvement in the child's life as a critical factor for engagement with violent extremism in Africa [4]. Other studies have explored how family cohesion and positive parent-child relationships could be a source of resilience in crisis settings [18–21] and therefore a possible deterrent for recruitment, while an emotionally supportive family environment can curtail retention in armed groups and positively influence reintegration [22,23]. Nevertheless, a deep understanding of the experiences of families living in conflict settings and how these experiences might influence armed recruitment is lacking.

To address these gaps in knowledge around the drivers of voluntary recruitment and how conflict and engagement in armed groups affects dynamics at the family level, this study comprised in-depth qualitative research with adolescents and caregivers in conflict-affected families in two distinct settings: Ouham-Pendé, Central African Republic (CAR) and North Kivu, Democratic Republic of Congo (DRC). The specific objectives of the study were to gain a greater understanding on the drivers of voluntary recruitment from the perspectives of adolescents and their caregivers, as well as to explore barriers and facilitators for reintegration of children associated with armed forces and armed groups. By better understanding the experiences and needs of children enduring recruitment and their families, practitioners can develop more comprehensive programming models to more adequately prevent voluntary recruitment and promote successful reintegration.

## Study setting

**Central African Republic context.** Research activities were conducted in three communities in the prefecture of Ouham-Pendé, CAR, which has been ravaged by fluctuating conflict since 2005. CAR has experienced waves of conflict since gaining independence from France, which colonized the country until 1960. Tensions between religious and ethnic groups within the region have driven civil conflict, with Northeastern communities (Muslim by faith) considered marginalized by the State and pro-government anti-balaka groups (Christian by faith) due to a lack of basic social services and infrastructure in the region and challenges to their citizenship. This escalated to the height of the military coup in 2012, and the country has faced multiple coup attempts in the last decade [24,25]. Communities on both sides of the conflict have suffered traumatic, conflict-related events, as well as exacerbated socioeconomic conditions and lack of governance [26]. The recruitment of children into armed groups has been ongoing—and increasing—in CAR since 2007 [27]. In CAR, three main non-state groups are listed in the Secretary General's report as parties committing grave violations against children, including child recruitment: Anti-balaka, former Séléka coalition and associated groups, and the Lord's Resistance Army. These and other armed groups on both sides of the conflict recruit children from their communities and use them as combatants, porters, informants, cooks, and for sexual purposes [24].

The government of CAR, along with the humanitarian community, has made strides to address the issue of children associated with armed forces and armed groups. In January 2016, the Government released the "Stratégie Nationale pour la Réinsertion à Base Communautaire des Enfants ex-Associés aux Forces et Groupes Armés en République Centrafricaine (national strategy for community reintegration of children formerly associated with armed forces and groups), which guides how humanitarian actors including the UN approach the reintegration of children in the community. This was followed by the adoption of the Child Protection Code in 2020 which criminalizes child recruitment and considers children primarily as victims. However, internal displacement and economic insecurity continue to trigger additional youth engagement, and persistent stigma against formerly associated children remains a barrier to their reintegration [27]. Further economic insecurity resulting from COVID-19 related measures have advanced destabilization throughout the country.

**Democratic Republic of Congo context.** Research activities were conducted in five research sites in North Kivu, DRC. Eastern DRC has been impacted by conflict for decades, first throughout the struggle for independence from Belgium which colonized the region for almost a century, and then enduring regional and internal political and refugee crises. These intermittent conflicts, combined with disease epidemics, food insecurity, and lack of infrastructure, have severely impacted communities throughout North Kivu. An estimated 5.2 million people have been internally displaced, 2.7 million of which are children, and more have fled to neighboring countries [28]. The recruitment of children into armed groups has been ongoing in DRC since both the First Congo War in 1996 and the Second Congo War from 1998–2003 [29]. In DRC, up to 30,000 children have been recruited and used in armed conflict since the beginning of the conflict [30]. Such recruitment continues today with 14 armed actors listed as committing grave violations against children, including armed groups such as the Mai Mai rebel groups in North Kivu, where the present study took place [31].

Government-led armed forces stopped enlisting children in 2003, and the DRC government signed the Agreement to End Child Recruitment and Other Conflict-Related Violations against Children in 2012, putting in place an action plan toward the release and reintegration of children. Nevertheless, thousands of children are still engaged with armed groups and child recruitment has increased since 2018 [24]. Similar to CAR, the economic impacts of COVID-19 have been linked to further economic and security destabilization throughout the eastern DRC.

## Methods

In-depth qualitative interviews were conducted with adolescents and caregivers across eight research sites in conflict-affected communities in Ouham-Pendé, CAR and North Kivu, DRC, where child recruitment is common.

### Study participants

To better understand the experiences of adolescents and their families living in conflict settings, in-depth interviews were conducted with adolescents currently or formerly engaged in armed groups or at risk of joining and their caregivers. Inclusion criteria for the study were adolescents aged 8–20 years who were either currently or formerly involved in armed groups or were determined to be at risk of joining and speakers of Swahili, Kinyarwanda, Sango, or French languages. In accordance with definitions set out in international law, association with armed groups could comprise any engagement including but not limited to living with the armed groups, engaging in combat, and conducting other tasks such as cooking or cleaning, as well as living at home and traveling to the group's base to trade with or sell goods to them. Adolescents who had not been engaged with armed groups were determined to be at risk based on socioeconomic criteria used by programmatic teams to identify and support children and young people in the community and prevent their recruitment by armed groups. Older adolescents ages 18–20 were included so as to gather information about their experiences when recruited as adolescents under the age of 18, and because this age group includes young people within the transitional developmental stage of adolescence.

In-depth interviews were also conducted with caregivers of adolescents who were formerly associated with armed groups to understand their perspectives on the drivers of child recruitment and the consequences of such engagement. Inclusion criteria for caregivers included being 18 years or older and being the caregiver of a child currently or formerly engaged in an armed group and speakers of Swahili or Kinyarwanda languages. A caregiver was defined to include any adult individual with whom the child was living who was responsible for the daily support of the child. Caregivers did not have to be the caregiver of a child participating in the research to also be included in the study.

Sites for the study were selected based on the presence of child protection case management services. In both countries, International Rescue Committee identified study participants through its child reintegration and psychosocial case management portfolio, which includes working closely with pre-existing community-based child protection committees in which community members trained on Child Protection are engaged by partners to support identification and referrals of children reintegrating from armed groups. Through community awareness raising led by these community-based networks, some children self-identify as formerly associated with armed groups and seek support, while others are referred by the community. Research participants in Ouham-Pendé and North Kivu were identified for the research through these community-based networks and informed about the study by the psychosocial counselors and child protection caseworkers who worked within the study sites and therefore had established rapport with the families in the community. Adolescents and caregivers who met the study criteria and demonstrated interest in participating were directly introduced to the research team to further explain the purpose of the study and obtain their consent to participate.

Purposive sampling was used to recruit study participants. Initial sample sizes were decided based on the caseload of the child protection program teams in each site, with prioritization of adolescents formerly engaged with armed groups and older adolescents, given the aims of the study. However, after some initial challenges identifying adolescent girl respondents and at-

risk respondents during data collection in CAR that inhibited reaching data saturation, the sampling approach for adolescents was altered and increased based on learning from this process. This involved determining the target sample size by calculating approximately 10% of the verified cases of child recruitment in each region. Verified cases are those cases that are verified by the UN as part of their monitoring and reporting mechanism (MRM), through which teams on the ground collect information and respond to grave violations against children. According to the UN General Assembly Security Council resolution 2427 [3], 442 children were recruited by armed groups in North Kivu, DRC (target N = 45 adolescents). While the number of children recruited by armed groups was not reported by prefecture in CAR, there were 310 cases of conflict-related grave violations in Ouham-Pendé (target N = 35 adolescents) in 2018. As a result, additional interviews were conducted in CAR. To compensate for the difficulties identifying adolescent girls to participate in the study in CAR, girls were oversampled in DRC to ensure adequate representation of their experiences in the study results. This enabled a more strategic sampling approach and the saturation of themes across research sites in both countries. We aimed to sample approximately 50% of the caregivers of adolescents participating in the study, with a target sample of 20 caregivers in each site and equal representation of men and women respondents.

A total of 74 adolescents (39 in DRC and 35 CAR) from ages 14 to 20 years were interviewed for the study, with respondents being 18 years of age on average in CAR and 17 years of age on average in DRC. Adolescent respondents who were currently or formerly engaged with armed groups but were currently living in their households are categorized as "out of armed groups". A total of 39 caregivers (21 DRC and 18 CAR) of adolescents currently or formerly involved in armed groups were also interviewed (Table 1) ranging from ages 32 to 66 years. Caregiver respondents were 50 years of age on average in CAR and 43 years of age on average in DRC. The majority of caregivers in CAR had no formal education and were earning an income through agriculture, trade, or small business. Similarly, the majority of caregivers in DRC had either no formal education or some primary education and were earning an income through agriculture or small business. A detailed, gender-stratified breakdown of participants can be found in Table 1.

## Data collection

The research applied a narrative inquiry approach in data collection and analysis, which aims to capture respondents' lived experiences and present a shared story which depicts themes and patterns based on the meaning and understanding that individuals give them [32]. This methodology was employed to intentionally focus on the stories of conflict-affected families and their strategies of coping and forms of resistance told in their own words. Semi-structured interview guides were utilized to guide conversations with adolescent and caregiver participants (see the guides in S1 Appendix). For the adolescent interview guide, separate interview

**Table 1. Formative research participants in the Central African Republic and Democratic Republic of the Congo by gender.**

| | Central African Republic (N = 53) | | Democratic Republic of the Congo (N = 60) | |
|---|---|---|---|---|
| | **Male** | **Female** | **Male** | **Female** |
| **Adolescents** | **27** | **8** | **17** | **22** |
| *Out of armed groups* | *24* | *5* | *13* | *15* |
| *At risk* | *3* | *3* | *4* | *7* |
| **Parents / Caregivers** | **10** | **8** | **8** | **13** |

guides were developed for adolescents who had previously engaged with armed groups and adolescents who had not engaged but were living in a community where recruitment was common, so as to ask questions that were specifically relevant to the experiences of those distinct groups of adolescents. To facilitate participatory conversations with the adolescents, the research team employed a narrative life-line technique. This approach visualizes the timeframe in which the participants were asked to recall information throughout the interview by drawing a timeline of their life while asking questions from the interview guide by period in the conflict: pre-recruitment, recruitment, and post-recruitment. The life-line technique is a recommended best practice for qualitative research on past experiences of violence [33], and has been used for research on conflict experiences [34] and research with children [35]. The guide included questions on their household dynamics and experiences during each of these periods and how they changed over time, as well as why they did or did not engage with armed groups and how they reintegrated into their families and communities. For caregivers, traditional semi-structured interview guides were used to understand caregivers' perceptions of their children's experiences during these periods, as well as their own experiences and challenges during the same time. Demographic information including age, gender, disability level, displacement status, marital status, and household composition were also collected from all respondents.

In-depth interviews were conducted with adolescents and caregivers across three sites in Ouham-Pendé, CAR in March 2020 and five sites in North Kivu, DRC in October 2020. Interviews were conducted one-on-one with the research assistant and respondent in Sango (CAR), Kinyarwanda, or Swahili (DRC) and audio recorded, with interviews lasting approximately 30 to 45 minutes in duration. Interviews were conducted either in a designated programmatic space or in a separate safe space of the respondent's choosing. Audio recordings were then transcribed and translated into French and English by transcribers and reviewed and verified by the research assistants who conducted the interview, as well as the data collection supervisors. The completed life line diagrams were photocopied and translated into French and English.

## Data analysis

Data were analyzed and presented descriptively utilizing narrative inquiry to depict the patterns in the shared stories of adolescents and families experiencing armed recruitment in terms of their narrative structure and content. Analysis was completed in two phases: 1) first analyzing data from each country separately, then 2) synthesizing findings and identifying common themes and distinct differences between each country. In the first phase, transcripts and photographs of the life-lines were coded and analyzed using the qualitative analysis software Dedoose. Data were coded using combination of *a priori* and inductive coding. An initial coding structure was developed using an adapted chrono-ecological framework [36,37], which examined the factors that influenced adolescent's recruitment and reintegration at different levels of the ecological model (institutional, societal, community, interpersonal, and individual) over time (pre-recruitment, recruitment, and post-recruitment). Using inductive coding, new codes were then added based on emergent themes and patterns in participants' responses for each of these levels and time points within the chrono-ecological model. Analyses focused on variation by demographic characteristics, examining differences in emergent themes by gender, age, and other characteristics. Detailed coding trees were then drafted with the combined deductive and inductive codes for each concept explored in the data and for each type of respondent, and a codebook was developed based on the coding tree. The coding tree and codebook were then reviewed by the research and programmatic teams and additional codes and definitions were added based on their feedback. Two members of the research team

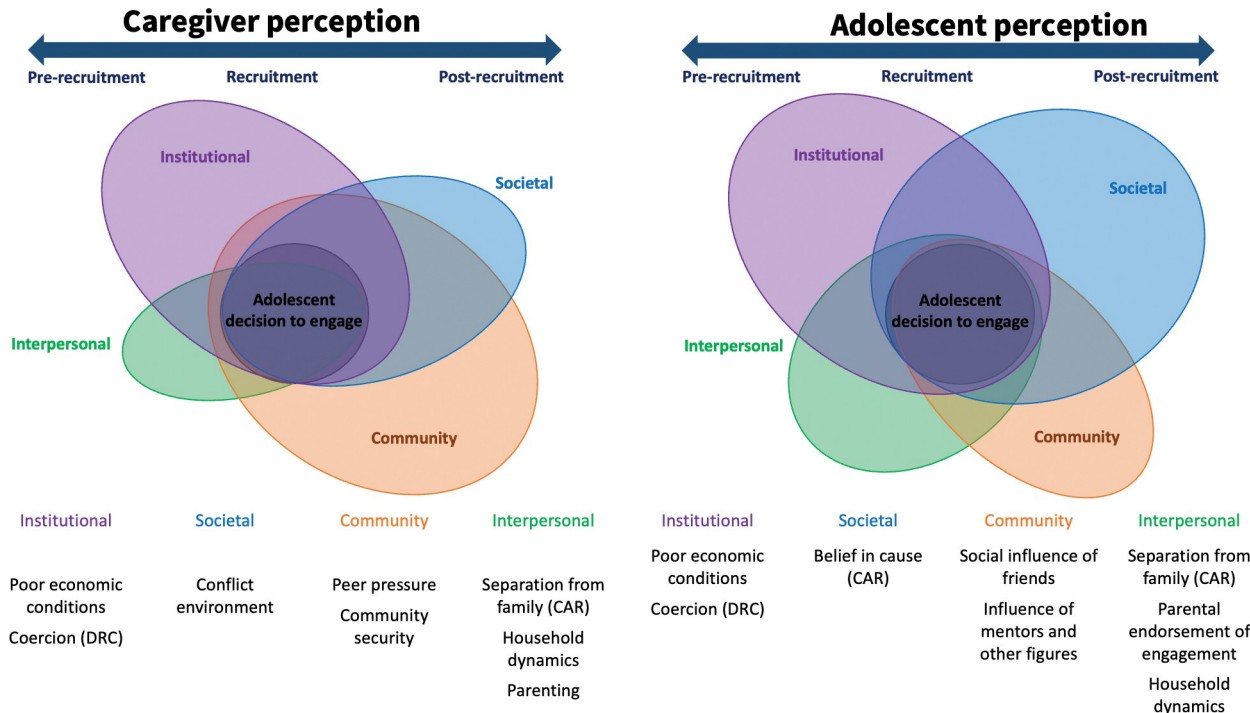

**Fig 1. A chrono-socioecological perspective of the drivers of child engagement with armed groups as reported by adolescents and caregivers.**

conducted the coding of transcripts, with double-coding conducted on a random 5% of the articles. After an initial round of coding, the study team met to review the inductive codes and discuss and resolve any discrepancies and coding criteria was adjusted.

During the first phase of analysis, separate validation workshops were held with the CAR and DRC programmatic teams (one for each country) including staff, caseworkers, and research team members from all research sites. During the workshops, the research team led discussions on the codebook, initial findings from that country, and points of clarification, and obtained crucial feedback for contextualization. In the second phase of analysis, the research team completed a second round of coding based on feedback from the country-specific validation workshops, which included adding several context-specific codes from both locations and re-coding excerpts. The research team then synthesized findings from both countries, and re-presented the findings and the resulting visual framework to the programmatic teams in both countries, as well as the external technical advisory group for final validation. Perceived importance of risk and protective factors for recruitment and barriers and facilitators of reintegration were determined based on the frequency of factors mentioned by adolescent and caregiver respondents. These have been graphically represented in the visual framework in Fig 1, with the size of each level denoting its frequency of occurrence in the transcripts relative to other levels. Quotations from participants are used throughout the presentation of findings to exemplify prominent patterns in the analysis.

## Ethical and safety considerations

The research protocol for this study was reviewed and approved by the International Rescue Committee institutional review board (CYPD 1.00.008 and CYPD 1.00.012) and the Comite National D'Ethique de la Sante (CNES) in DRC. All research methods and tools were also

reviewed by an external technical advisory group (TAG) convened for the project, which was made up of eight global, regional, and local experts on research and programs for children associated with armed forces and armed groups, including representatives of academic institutions, UN organizations, and international and national governmental organizations and service providers. The TAG met periodically throughout the research design and implementation and provided feedback on research tools and methodological and ethical procedures, which were adapted based on their suggestions. In addition to the TAG, the protocol and tools were reviewed by local service providers and community-based child protection networks who provided inputs and advised on contextually relevant risk mitigation procedures, measures for ensuring confidentiality and privacy, informed consent, and referral procedures.

All participants gave their verbal informed consent prior to their participation, including detailing that data would be published with no identifying information. Ensuring the safety of the respondents and the research team was the top priority during and after data collection. The research adhered to guidance for ethical research on sensitive subjects such as UNICEF's *Ethical Research Involving Children (ERIC) project compendium* [38], the Population Council's *Ethical Approaches to Gathering Information from Children and Adolescents in International Settings*: *Guidelines and Resources* [39], as well as WHO's *Ethical and Safety Recommendations for Researching, Documenting and Monitoring Sexual Violence in Emergencies* [40]. Informed consent and assent were obtained verbally from all participants. Trained data collection staff administered informed consent of the caregiver for their own participation or parental consent for the participation of their child in those cases where the adolescent was less than 18 years of age. If parental consent was given, the data collection staff then administered informed assent of the adolescent for their participation. For those adolescents ages 18–20, informed consent was administered directly to the youth by the data collection staff. All consent and assent procedures and interviews were conducted in private spaces. Research assistants reviewed the informed consent form with all participants before obtaining their consent and made substantial efforts to ensure that participants understood their decision would have no influence on their ability to receive services. Referral information was offered to all participants and those adolescents who were not already engaged in child protection programs were referred directly by a caseworker for individualized support.

Gender-matched data collection staff were hired locally and were screened for prior experience in child protection and/or psychosocial counseling for children. Research assistants received training and practice in maintaining ethical principles such as ensuring privacy, confidentiality, and safety of participants, as well as working with children and recognizing verbal and non-verbal cues to anticipate their needs. Research assistants also received training on reflexivity and bias, including practice-based roleplaying and piloting of the research tools with child protection staff who are experienced with working with children in conflict settings. A data collection supervisor with prior experience working on research on child wellbeing accompanied each research team during data collection and child protection caseworkers were on-hand for direct referral and to respond in any adverse situations in all sites.

Participants were assigned an ID number when enrolled in the study, which was included on the informed consent forms and used to link audio recordings, transcripts, and any other documents from the interviews. Data files including audio files, transcripts, and scanned documents were saved in password-protected files on locked computers held by the research managers in each site. Once audio recordings were saved to the computer, they were deleted from the audio recording device. Participant ID, gender, age, and village were included on transcripts, but all personally identifying information, including the names of specific towns and armed groups, were removed from transcripts, notes, and drawings during transcription. In consultation with local partners, presentation of analyses removed any other information that

could lead to a breach in confidentiality, such as the name of the research site, including quotes within this paper.

## Results

This paper examines the perceptions of adolescents and their caregivers on the factors that influence them to engage with armed groups, as well as their experiences with the groups and upon returning home to their families and communities.

The qualitative results illustrate that families living in conflict settings are subject to traumatic experiences and economic hardship that can further erode protective family relationships. This leaves adolescent boys and girls particularly vulnerable to the systemic and overlapping interpersonal, community, social, and institutional push and pull factors that might influence them to engage with an armed group. These drivers are cyclical, preventing adolescent boys and girls from reintegrating into their families and communities and contributing to the common but difficult decision to return to the armed group. Key findings for adolescents and caregivers are summarized by country in Supporting Information, S2 Appendix. The findings are presented below using a chrono-ecological framework to reflect the cyclical nature of the different factors that influence adolescents' experiences over time and at different socioecological levels. To illuminate dynamics within the household and how a child's engagement with armed forces and armed groups affects the larger family environment and the interpersonal relationship between the child and their caregivers, adolescent and caregiver findings are presented together within each section.

### Pre-recruitment

**Institutional level: Economic insecurity and its influence on the household.**   Economic drivers were reported by boys and girls in CAR and DRC; the majority of respondents reported that their families' livelihoods were destroyed by the conflict, and they had no access to basic needs, leaving them with few options. Adolescent boys in particular described a sense of having "nothing to do" before they joined. One 20-year-old adolescent boy from CAR stated,

> *"I lived together with my mother, then I moved to be together with my friends because there was nothing to eat at home, and also finding work to do in the village had become difficult. . . . If I go [with the group], I would find money to start my business again and this is how I accepted."*

The conflict situations in both CAR and DRC were further exacerbated by the economic consequences of COVID-19 and related restrictions on movement. One 15-year-old girl from DRC expressed:

> *"When I learned that schools were closing due to COVID-19, I went straight into the forest. . . . I lived with my friends. We went [to where the armed groups live] together. . . . After whipping us, they gave us money to buy food for them. . . . [My parents] could accept because of the lack of money."*

Male and female caregivers in both countries also described being unable to adequately provide for their children due to destruction of their livelihoods and homes during the conflict. With no income and no ability to provide for their children's basic needs, send them to school, or help them get a job of their own, several caregivers expressed that their child had no choice

but to join the armed group. Male and female caregivers in DRC particularly emphasized these economic drivers as the main factor that drove their child to engage with the group. One male caregiver of an adolescent boy in DRC described:

> "What motivated him, it was because of poverty. He could see there is nothing at home, so he decides to join. . . . I talked to him asking: what pushed you to go there? He told me he realized that life conditions are becoming bad in the village. He heard the armed groups had something good they would provide to improve their lives."

Caregivers emphasized that the economic and social drivers that influenced their child left them with "no choice", even though they as caregivers had asked them not to go. This vulnerable economic situation demonstrates the difficulty of distinguishing between forced and voluntary recruitment for families living in situations of armed conflict. In both DRC and CAR, adolescents described their "choice" to engage with the groups while also describing situations of economic exploitation and violence. Two girls in CAR reported being kidnapped and forced to join the armed group while simultaneously feeling motivated by the cause to "join" the group even though they were forced to join. In DRC, several caregivers described their children experiencing coercion that verged on forced recruitment while still reporting that their child joined voluntarily. This was most often reported by caregivers of adolescent girls compared to adolescent boys, such as one mother of an adolescent girl in DRC who said, "The reason is that life is difficult here and many get to join after having gone to the bush to carry embers. . . . She went to look for money without knowing she was going to be kept there".

**Societal level: Conflict environment and insecurity.**   Adolescent boys and girls in both CAR and DRC described a childhood mired by conflict, loss, and displacement. All respondents in both countries described experiencing multiple traumatic conflict-related events, including attacks on their village, flight and displacement, kidnapping, killing of family members, and separation from their families. In CAR, these experiences informed their decision to join the armed groups, as the majority of both boys and girls described their desire to avenge family members and a personal affiliation with one of the militias as the primary motivating factor for engaging with a group. A 20-year-old boy from CAR described, "Yes, the [opposing armed group] killed my two older brothers in [village]. . . . Which made me angry and decide to join the armed group to avenge my two older brothers".

A few boys and girls interviewed in CAR were also separated from their families while fleeing attacks on their village and reported having no way to support themselves and nowhere to go other than to join the armed groups. This differed from adolescents in the DRC, as the majority of adolescent respondents described not liking or supporting the group, and none referenced avenging family members, defending their community, or other motivating factors related to the cause of the groups. Indeed, some adolescents explicitly stated that they did not trust the armed groups but still engaged with them for economic reasons. For example, one 15-year-old girl from DRC who reported that she went into the forest to join with the group to earn money shared: "No [I did not believe in their cause], I didn't trust them. . . . Because they are bad people. Anyone who holds a weapon is very bad and dangerous. A person with a gun is a very dangerous person".

Interviews with caregivers of children who were recruited by armed groups also shed light on the situation of families during their child's recruitment. Similar to adolescents, caregivers in both DRC and CAR reported having experienced multiple traumatic conflict experiences, including their children being killed or separated from them during displacement. Several caregivers in CAR reported that their child joined the armed group during this period of separation and not knowing their whereabouts, while several caregivers in DRC described assuming their child had been killed until learning they had joined the armed group.

**Community level: The influence of social networks.** While adolescent respondents did not explicitly list their social network as a key factor in their decision to join, most described the overall influence of local militias on their families and communities. In both CAR and DRC, adolescent boys and girls reported deciding to engage with armed groups alongside friends or other community members. The influence of these networks were distinct when comparing the experiences of adolescents who were associated with armed groups to those who were not. Several at-risk boys in CAR and DRC reported that male community members and extended family members specifically advised them against engaging with the group. At-risk adolescent girls and boys reported not having any friends in the group, further reducing any social influence to engage. Among those adolescent girls and boys who engaged with the armed groups, none reported that anyone from the community discouraged them from joining. Rather, boys described having support from their friends and members of the community for joining and having many friends in the group, though this was reported less often among girls. One 19-year old boy respondent from CAR described the social pressure to join the armed groups, and his fear of not conforming:

*"Once at the beginning of last year, I took a gun from a friend's house, they were passing in a group. I said to him, my friend, you have already entered the armed group, but I, my parents refused to enter it, can you give me your gun so that I can hold on a little? . . . My heart was beating hard, I thought to myself that I had to go back there, so that we looked alike. If I did not enter the armed group, I would always be inferior to the others, there will be a big difference between them and me."*

This community-wide pressure to join among youth was a driving source of motivation reported in both CAR and DRC.

**Interpersonal level: Gender differences in parental endorsement for engagement with armed groups.** Respondents who were associated with armed groups reported varying levels of support from their parents. Both boys and girls had family members who were also in the group, such as fathers or siblings. Girls out of the armed groups reported more often than boys that their parent gave them permission to join. A few girls who were at risk but who had not engaged with armed groups reported that their caregivers had forbidden them from joining or that they could not join because of their gender. One 16-year-old girl from CAR described how she wanted to join but was not allowed: "As I am a girl, I could not even avenge my brothers. Right now, I can't run behind men." Still, several girls from both CAR and DRC reported engaging with the group despite their parents' disapproval.

In contrast, several boys in both DRC and CAR reported that their parents were against them joining but that they made the decision to run away and engage with the group regardless, creating tension with their caregivers. Child protection counselors in CAR illuminated this dynamic between parents and their children, explaining that armed groups pass through the community and ask what each family will contribute. Even if parents do not want their child to go, the contribution of their child's labor guarantees security for the family. In the case of boys, they were not expected to ask for permission; as was reported in both DRC and CAR, many caregivers did not know their boy child had joined until he did not come home. In the case of girls, given gender roles and decision-making dynamics within the home, they would need to have the support of their parents—particularly their fathers—to engage with the group. In addition, any economic benefits obtained by the girl in the group would come back to the family, whereas if an adolescent boy has economic gains, they might use them to start their own household. This dynamic was also present in the DRC, where several adolescent

girls reported that their caregivers supported their engagement in the group for economic gain, even if they were concerned about their safety. One 15-year-old girl described:

> "My parents allowed me when I was growing up to go into the forest to associate with armed groups. . . . So that I can have the money. . . . And after I earn some money, that I can go home to help them pay for clothes, for example. And the money they allegedly used to buy me clothes would pay for school fees. . . . They don't like [armed groups] . . . because they have arms, and they can kill people."

Many male and female caregivers of both boys and girls reported being vocally against their child engaging with the group but that their child still made the decision to go. This was mostly reported among caregivers of boy children; among caregivers of girl children, several specifically stated that their daughters did not engage with the armed group because they did not allow them to.

Similar to the protective influence of community members, most of the at-risk boys who had not joined armed groups reported that their parents were vocally against them joining.

**Interpersonal level: Household dynamics as a push or pull factor.** Several adolescent respondents reported that the negative situation with their family in their home—for example, adolescents described both verbal and physical abuse, as well as a general unsupportive family dynamic and feeling unhappy or unwelcome—drove them to affiliate with the armed group. This was reported mostly by adolescent girls in the DRC. One 17-year-old girl in the DRC shared about her relationship with her parents before she engaged with the group: "It was negative because, when one goes wandering, it is not because there is nothing to do at home. There are chores at home, but you go. If they are pouring out abuse on you, you decide to go wandering." In contrast to those who had joined armed groups, none of the at-risk youth in both CAR or DRC who had not engaged with armed groups reported having negative relationships with their parents before or after the conflict.

Family responsibility may also be a protective factor for preventing child recruitment, particularly for older adolescents. In both countries, older at-risk adolescent boys and girls who reported being financially responsible for their families described having to take care of their younger siblings or mother and therefore being unable to leave home to engage with the armed groups.

## Recruitment

**Societal level: The influence of social norms.** Both boys and girls described a normalization of violence within the groups, such as being whipped (among girls) and beating each other as punishment or "for fun" (among boys). Several boys and girls interviewed described poor living conditions in the camp and not having access to basic needs such as food, water and personal hygiene. Very few adolescent respondents reported being paid or receiving any economic or material compensation, despite many having engaged with the group for this reason. Several boys and girls in both CAR and DRC expressed regretting their decision to join once they arrived and feeling alone and unsupported by adults. Despite this violent environment, a few older adolescent girls in the DRC reported staying in the group because the situation there was better or the same than the situation they would return to at home. One 18-year-old girl in the DRC explained that she stayed for three years with the group because she perceived her situation at home to be the same:

> "What scared me was that I realized that the life I led in the group was the same as at home. That's what made me decide to go home so I could die next to my parents. . . . I [stayed 3 years] because I knew that even with us life was difficult."

**Interpersonal level: The influence of family contact and support while with the armed group.** In CAR, the majority of adolescent boys reported living with the group and the majority of adolescent girls lived at home, with exception of a few who lived with the group, two of whom were separated from their families. Those who did live with the group reported visiting their families while in the group. In DRC, both boys and girls reported living with the armed groups and most reported being unable to visit home or keep in contact with their families during that time.

After their child joined the armed group, the majority of caregivers in CAR reported having little ability to see or communicate with them and therefore had a limited understanding of their experience, though female caregivers reported more knowledge of their child's time in the group, in general. In the DRC, both male and female caregivers more often reported having contact with their children, but most often male children. In contrast, caregivers of girl children in the DRC reported learning about their daughters' experiences after they returned, but had no contact while the girls were in the group. Among those caregivers who did know about their child's experience, they described the violent and poor living conditions of the encampments. Several reported that these poor living conditions drove their child to leave the group. One male caregiver of an adolescent boy in CAR said:

> *"There, it was just suffering; here at home, he slept under the blanket, but there, he slept in the open space. At some point, he couldn't bear it and he was obliged to come here to the house to take the blanket."*

Across age groups and gender, adolescents who were in contact with their families during the conflict reported that their caregivers did not want them to be in the groups once they joined, and many expressed that this disapproval influenced them to eventually leave the group. A 13-year old girl from DRC explained:

> *"I preferred to do a week in the group because I had no idea to go home. . . . I thought I couldn't adapt to life at home. . . . There was a message that my family sent me when I was in the group and it gave me the courage to go home. I used to think I couldn't be accommodated anymore."*

Adolescent boys and girls in both countries described their caregivers' desire for them to return home as one of the main motivating factors to leave the groups.

Similarly, most male and female caregivers reported that their children made the decision to leave the armed groups and return home because they continually expressed concern and asked them to come home every time their child visited. Caregivers described having this conversation in different ways, with some telling their children they were needed at home, and others advising them that they could make a more stable income if they left the group. One female caregiver of an adolescent boy in CAR shared how she discussed with him about different opportunities for income:

> *"I approached him with good advice. I asked him to stop going to the armed groups. . . . I advised him to look for other activities, even if it involves selling firewood. . . . I'm afraid he'll die, I won't see him again, because the armed group is not for children and a lot of bad things can happen there."*

Another female caregiver of an adolescent boy in DRC reported talking to her son about his future to motivate him to disengage:

*"I discussed [his experience] with him . . . to know what he was doing there. . . . We discussed how he can leave the armed groups and come back home. . . . Not going there again so that he cannot die there and to avoid to make again alliance with those guys. . . . I gave him pieces of advice. . . . I told him I am going to get him back at school. . . . When I talked about this he showed joy and it gave him courage."*

The perception of caregivers that their support influenced their child to leave the group is consistent with what adolescents reported above and demonstrates the effect the caregivers still had on the child during their time with the group. Most adolescent boys and girls described leaving the groups because the immediate conflict ended, and their caregivers asked them to come home. Older adolescents also reported needing an income to support their families and wanting a stable future that being part of the armed groups could not provide.

Nevertheless, many adolescent boys and girls in both countries also described positive social dynamics within the armed groups, including receiving social support from other boys and girls in the group, as well as protection and mentorship provided by chiefs or leaders. A few separated male and female adolescents in CAR described this support as filling a need they had after losing their parents in the conflict. One 20-year old boy from CAR said, "As I lost parents, my friends from the armed groups supported me when I was among them. When someone wants to provoke me, they often defend me. . . . When I miss a few things, they support me."

**Interpersonal level: Negative household dynamics resulting from engagement with armed groups.** In the DRC, caregivers emphasized the consequences of their child's engagement for the entire household. Both male and female respondents commonly reported experiencing stigma from the community and their relatives who blamed them for their child's engagement with the armed group. In at least three cases, a caregiver within the household was imprisoned or officially punished by authorities for their child's engagement. One male caregiver of an adolescent boy in DRC described,

*"What we experienced when the child was in the armed group, firstly the government took us to prison. We were charged to pay fines and were obliged to find this child, get him leave the armed group and bring him back home."*

While this stigma was reported by both men and women, female caregivers of girls and boys reported receiving additional blame from their husbands due to the perception that it is the mother's fault when their child misbehaves. In some cases, women reported that their child's engagement increased tension within the household, including fighting with her partner. One female caregiver of an adolescent girl expressed this worsened dynamic: "Our relationship was bad when the child joined armed groups because [my husband] was saying I was the cause of her joining armed groups." This increase in tension was reported by female caregivers of boys and girls, and respondents reported that it contributed to their own struggles with mental health. Male caregivers also described a toll on their mental health as a result of the stigma they experienced in the community from their child's involvement.

## Post-recruitment

**Institutional level: Continued economic insecurity impedes reintegration and influences household dynamics.** Upon leaving the armed groups, adolescent boys and girls in both DRC and CAR returned to similar conditions in their homes and communities as before they left. Many described having no access to basic needs and a lack of employment opportunities, and their household having to rely on community financial support to survive. Caregivers

in both countries described a similar situation upon their child's return home: both male and female caregivers described being unable to adequately provide for their children, which they expressed drove their children away. They also reported that the inability to find employment was a risk for adolescents to rejoin armed groups. One female caregiver in CAR who had multiple children join armed groups described:

> "We are really concerned about the future of our children. When your child is out in the armed group and you have nothing to give him, it hurts. Me and my husband have a lot of worries because if we cannot find something for these children, they will one day risk re-entering the armed group. When they ask us something, we are unable to give them, it hurts as parents."

This lack of financial stability and inability to find jobs was described by both male and female caregivers in CAR and DRC as a source of tension within the household.

Caregivers also commonly listed economic assistance as a critical need for supporting their families and preventing re-recruitment of their children in the future. Male and female caregivers in both countries most commonly reported needing livelihoods training for their children or themselves in order to have a more sustainable income. This included needing financial or material capital to set up businesses in the face of economic hardship in their communities and a lack of available livelihoods opportunities.

**Community level: Gendered experiences of reintegration and stigma for girls and boys.** Adolescent girls in both CAR and DRC reported experiencing gender-based stigmatization from their families and communities for having spent time with armed men. A 15-year-old girl from CAR explained that other girls treated her differently: "Since I came back, I have lived in perfect harmony with my little brothers. But some of my fellow girls call me a brothel because I sold porridge to the [armed group]. However, others continue to play with me." Similar instances were reported by adolescent girls in DRC, with one 17-year-old girl sharing: "With my brothers, we love each other very much. . . . The girls remind me of what happened." This gender-based stigma was acknowledged by female caregivers who described the discriminatory treatment their girl children experienced at the hands of other family and community members for their association with armed men. In contrast, only a few boys mentioned experiencing some stigmatization in the community upon first returning from the groups, which they said lessened with time.

A few adolescent boys in both CAR and DRC expressed that they could not or did not want to speak with their friends families about what had happened to them. They described this as a challenge that made them less likely to spend time with their families. However, other boys in both countries described having a positive experiencing speaking to their caregivers about their experience. One 19-year boy in CAR said:

> "They asked me if I kept remembering what I did during the crisis, I replied that I forgot everything, I don't remember anything anymore. Then I told them not to ask me that kind of question. I don't like to go back to what happened."

**Interpersonal level: Adjusting to life at home and reintegration challenges within the household.** While many caregivers described positive family relationships upon their child's return, others described tension within the household and challenges with the change in behavior of their child. These challenges were described mostly among caregivers who had boys in the armed groups, and less among those with girls. Caregivers described their boy children struggling with anger and aggression, and a general disinterest in wanting to occupy

themselves with school or work. One female caregiver in the DRC did report that her daughter's behavior had become more aggressive and the negative impact her behavior and the family's poor economic situation had on their relationship:

> "We are still not in a good relationship because all that she could receive from me she is not receiving. . . . We have nothing because she wants to study but we do not have the means to pay her school fees. . . . She loves her friends with whom she shares tobacco, drink and other drugs. But she doesn't consider anyone who wants to give her advice.. . . . When she is angry and I try to give her advice, she speaks unbearable language to me."

In a few cases, caregivers with girl children revealed their daughter's desire to gain autonomy after returning from the armed group. In these cases, their daughter's change in behavior was described negatively as an example of the influence of the armed group on her attitude. One male caregiver of an adolescent girl in DRC described this situation:

> "When she arrives home, she no longer wants to be considered a child. When she talks, she wants us to do everything she said. And if a family member including myself, talks to her in terms of her advice, she reacts very negatively saying that we have nothing to say to her. That if she wants, she can go back there. And about a few days later, she's going there. In short, she wants to have authority over everyone in the house and that there is no one else talking in the house."

In comparison, most adolescents reported that their family relationships went back to normal after they returned home, though for some it was more difficult. Several boys described the tension with their caregivers as they readjusted to being back at home. This tension often related to the feeling that they were not receiving adequate support from their families who could no longer provide for them. One 18-year-old boy from CAR explained:

> "After I left the armed group, my relationship with my caregivers here has not improved at all, they do not pay attention to what I do. First, I don't live with them anymore, I don't eat with them. . . . What hurts me today is the disagreement between me and my guardians here. It gives me a lot of negative thoughts."

**Interpersonal level: Changes in parenting and caregiver needs and support for the future.**   Caregivers also expressed their own challenges with parenting after their child returned home. In DRC and CAR, caregivers described their own mental health issues and the difficulties this caused with parenting and adequately supporting their children. Both male and female caregivers shared that the separation and displacement their families had experienced resulted in prolonged mental health effects that still made it more challenging for them to care for their children in present. Respondents described these mental health impacts lasting beyond their child's return from the armed group, and having a continued influence on their ability to care for their children. One male caregiver of an adolescent boy in who joined an armed group in CAR expressed:

> "I fought every night for my family to have enough to eat and watched them at all times so that something wouldn't happen to them. We had lost hope, I thought my life was already ruined, I lost even self-confidence, but it was God who guided me to deal with some of these problems."

However, many caregivers also remained optimistic about their relationship with their child and their child's future, and they described the approaches they had used to support their children which they perceived to be successful. Common strategies included talking to their children about opportunities for the future, sharing a treat or meal as an entry point to a conversation, giving them a responsibility, and talking together while working. One male respondent in CAR described trying to involve the whole family in supporting his son and having both individual and family conversations with him to create an atmosphere of communication and support. A female respondent in DRC described her approach to talking to her daughter while working with her:

> "Sometimes we sit down together and I take the opportunity to give her advice on how to live. . . . I do it when we are doing a given job because sometimes she agrees to go with me to the field. . . . When I'm with her, I ask her questions about life. I ask her what she would like to do so that we know what can help her in the days to come."

However, in both countries, while female caregivers reported supporting both boy and girl children through different approaches, male caregivers exclusively reported supporting their boy children. No male caregivers of girl children shared examples of positive approaches.

Despite having examples of successful approaches for talking to and supporting their children reintegrating, male and female caregivers in both countries expressed needing assistance to adequately support their families. In CAR, several male and female caregivers specifically mentioned wanting to increase their knowledge and skills on child development, child wellbeing, and how to practice better discipline. Male and female caregivers of both boy and girl children also described wanting to build skills on how to talk to their children about the conflict, in particular. One female caregiver of an adolescent boy in CAR requested: "You give good advice to parents so that we can help our children also through advice, so that they can put their heads out of the bad things of armed groups. . . . It's the advice which facilitates a child's well-being and development." Male and female caregivers in DRC also emphasized wanting to build their skills to talk to their children about the conflict. One male caregiver of an adolescent girl in DRC spoke about what he support he needs:

> "First, for me, it's building capacity, giving a lot of training so I can know how to advise this child and de-traumatize her. If I attend a lot of training seminars on how to live with the children or how to give them advice, it will already be a lot to help this child."

## Discussion

The study findings revealed the multiple and overlapping drivers of 'voluntary' child engagement with armed groups, many of which are also barriers to reintegration upon the child's return to their family and community. Fig 1 depicts the chrono-socioecological framework used to summarize the influence of the institutional, social, community, and interpersonal factors that influenced the individual child's engagement with armed groups and reintegration into the community before, during, and after the conflict. Building on existing socioecological models of children and families in conflict settings, this adapted model shows the drivers of recruitment from the perspectives of both the adolescent respondents and their caregivers at different chrono-socioecological levels. The model separates out the socioecological levels, with the size of each level denoting its relative importance as determined by occurrence in the transcripts. The time component is represented to emphasize the cyclical nature of conflict and recruitment in these settings and how these factors fluctuate throughout these periods of

an adolescent's experience. The separate depiction of adolescents and caregivers is not meant to contrast their perspectives but rather to draw out the central factors that both sets of respondents perceive to influence adolescents to engage with armed groups, and to understand the overlapping needs of families living in conflict settings that affect household dynamics and erode the protective family environment.

## Institutional level factors

Many of the drivers of child engagement with armed groups noted by adolescents and caregivers in CAR and DRC are common push and pull factors identified in other literature on child recruitment. For example, economic drivers such as community-level poverty, lack of employment and education opportunities, household economic instability, and monetary incentives or status provided by the group were frequently mentioned by respondents in both CAR and DRC. Evidence from South Sudan, CAR, DRC, and Colombia demonstrates that these economic drivers including those listed above are among the most common factors that influence recruitment across a number of different conflict and geographic contexts [13,41]. Disruption to economic activities also leaves many families dependent on the income they can receive from their child being engaged with an armed group [13,42]. Having alternative options for income generation has been found to be a protective factor against recruitment [13].

In DRC, coercion overlapped with economic push and pull factors, with caregivers describing their children being unable to say "no", so as to protect their economic and personal security. Even in some situations where adolescents were reported to have some agency in their decision (e.g. they reportedly "chose" to engage), caregivers and adolescents described situations of forced recruitment, kidnapping, and threats to their safety. While this study focused primarily on voluntary recruitment, the findings demonstrate the difficulty of distinguishing between circumstances of coercion and circumstances of agency. However, forced recruitment of children into armed forces and armed groups is widely documented across conflict contexts [13].

## Societal level factors

The social influence outside of the household through peer and community pressure also played a large role in motivating young people to engage with the armed groups. In CAR, adolescent boys and girls identified this social pressure as an affiliation with the cause of the armed groups. The ethnic and religious nature of the conflict in CAR could be why this social pressure at the community level was so influential. Both caregivers and adolescents described scenarios in which all of the young people moved to join with the armed groups at the outbreak of armed clashes, which occurred frequently and continuously. Aggressive recruitment tactics and propaganda by armed groups that leverage this social pressure and religious affiliation have been documented in other evidence from CAR (24). A sense of community expectations for vengeance has also been documented in the Kurdistan Republic of Iraq, DRC, Mali, and South Sudan [13,23,42,43].

## Community level factors

Social influence also played a role in the DRC where adolescents considered themselves much less affiliated with the values of the armed groups. Caregivers in DRC emphasized the influence that their children's friends had on their decision to engage, and adolescent boys and girls reported making the decision to both engage with and disengage from the armed group with friends and other adolescents in their community. Social pressure plays a significant role in recruitment as children seek a sense of belonging, particularly in situations where peers and friends have already joined an armed group [13,23,42]. Other literature from CAR and DRC

found that children wanted to be with friends who had already been recruited by armed groups. Similar camaraderie was found to influence young people joining armed groups in a number of different conflict contexts including South Sudan and Colombia [13].

## Interpersonal level factors

When examining social networks, this study focused particularly on the influence of family- and caregiver-child dynamics within the household. The findings revealed that these relationships play an important role in a child's decision-making. For at-risk adolescents who had not engaged in armed groups, respondents reported that their caregivers or other family members had explicitly advised them not to engage with armed groups and discussed with them possible consequences of this decision. None of these at-risk adolescents reported having a caregiver condone the armed groups, whereas adolescents formerly associated with armed groups reported having parental approval more often, though endorsement from caregivers was overall mixed. However, this parental influence had some gendered differences. Girls more often reported that they engaged with armed groups with parental support, as opposed to boys who did not report seeking or needing this approval. Girls who previously engaged with armed groups also more often reported having particularly negative and harmful relationships with their caregivers, including emotional and physical maltreatment and neglect, which drove them to leave home to engage with the armed group. Family violence has been identified as a driver of child recruitment for girls in other conflict contexts [42].

Nevertheless, in situations where adolescent boys and girls did engage with armed groups with or without parental condoning or approval, the role of the caregiver remained important to the child's decision-making. Several adolescent boys and girls reported staying in contact with their caregivers, valuing their opinion, and disliking the tension caused by their decision to engage with the group. Many adolescents in both DRC and CAR described this caregiver influence as being a main reason they decided to disengage from the group; or, in the case of a few girls with particularly negative home environments, the reason they decided not to disengage. Literature on child recruitment supports the finding that family support for the armed group, including being engaged in the armed group themselves, can be a pull factor for children to join [13,42]. More broadly, substantial evidence from across geographic settings demonstrates that parental behaviors and lack of family cohesion are associated with child maladjustment [44–48]. Similarly, the findings revealed that caregivers are experiencing significant mental health and parenting challenges as a result of their own conflict experiences, compounded with the additional tension, stigma, and other challenges faced by the household as a whole due to the child's engagement with armed groups. While there is little existing evidence on the specific experiences of parents or recruited children, recent studies have demonstrated that caregivers' own mental health issues are associated with neglect and harmful practices toward children [49–52], in turn negatively affecting child mental health and wellbeing [53,54]. This perceived lack of parental involvement in a child's life and child maladjustment can be risk factors for engagement with armed groups [4].

Literature on the potentially protective role of the family for child recruitment specifically is limited. However, recent studies across a variety of conflict contexts in Latin America and Southeast Asia demonstrate that parent-child relationships could be a crucial source of resilience for youth [18–20,55]. The present study also found that these relationships had an influence on the adolescent boys' and girls' decisions to disengage from the group. Evidence confirms that having a supportive family environment can undermine retention within armed groups and facilitate the successful reintegration of adolescents into the community [22,23]. Social support has also been found to mitigate the long-term effects of perpetrating violence

during war and post-conflict stigma for children [56]. These findings show the importance of a supportive home environment to insulate children from the drivers of recruitment and also to support their reintegration. Given gender dynamics within households where in some contexts the male caregiver is considered the head of household while the female caregiver is considered responsible for caring for and disciplining children, this supportive environment should involve all caregivers and trusted adults within a family. Recent studies from refugee and conflict-affected populations have demonstrated that parenting interventions can improve parenting efficacy and skills, family- and parent-child relationship outcomes, and reduce negative family interactions, harsh parenting, and subsequent child mental health and behavior issues [19,57–60]. Nevertheless, few parenting programs have focused specifically on the issue of recruitment into armed groups, and current parenting program models do not address the complexity of factors that affect parenting in conflict settings [54]. Qualitative data obtained from participants of a parenting program in North Kivu, DRC also drew attention to the specific and unaddressed needs of caregivers of adolescents who are reintegrating into their families after disassociating from armed groups [61]. While this study addresses a critical gap in evidence on household dynamics and parent-child relationships for families experiencing recruitment, more research is needed to examine how parenting programs might address these needs.

## Conclusion

The findings from this study are based on qualitative interviews from a small sample of adolescents and caregivers in two conflict settings in central Africa, and therefore not generalizable to all active conflict settings. While the sampling approach involved the recruitment of study participants through established community networks for Child Protection to ensure their safety and the safety of the larger community, this purposive sampling strategy may have resulted in sampling bias, as those adolescents who are connected to existing reintegration or psychosocial services likely have different experiences than those who are still engaged with armed groups and have not been able to access services. Similarly, the extreme level of unmet need in the research sites could have led to response bias if adolescent or caregiver participants felt their responses would gain them access to additional services or aid. While significant efforts were made to explain that their participation or responses would not affect their ability to receive services to ensure the voluntariness of participants, this is always a risk of research in humanitarian contexts.

   Finally, the analysis of the findings for this paper focused on the interpersonal level, examining the drivers and consequences of adolescent engagement with armed groups within the household and how drivers and other levels of the chrono-ecological framework also influence family dynamics. Nevertheless, the findings highlight the value of a family-based approach to improving adolescent resilience to recruitment by armed groups. More programs are needed to support caregivers in armed conflict settings to improve their own mental health and family functioning, and to better support their children to prevent recruitment, to facilitate their successful reintegration, and ultimately to reduce risk of re-engagement. Given the strong influence of family and parent-child relationships on the wellbeing of children—and the potential protective effects of these relationships against recruitment—parenting programs could be an effective approach for preventing the engagement of children in conflict. More research is also needed to fully explore household dynamics and the pathways through which these dynamics can be protective against child recruitment.

   Many of the drivers identified in this study cannot be addressed through interpersonal- or household-level approaches, alone. While the household unit—relationships with caregivers,

siblings, and extended family members—are potentially protective against child recruitment into armed forces and armed groups, some community- and institutional-level factors such as poverty, lack of infrastructure and lack of employment and education opportunities cannot be addressed through such an approach. Linking family-based programs to support from other sectors such as livelihoods or education programs targeting adolescents or systems strengthening for infrastructure and governance will be key to an integrated and comprehensive approach to preventing child recruitment into armed forces and armed groups.

## Supporting information

**S1 Appendix. Data collection instruments.** Guide A. Individual Interviews for Boy/Girl Children and Adolescents Life Line Tool: Previously engaged with armed groups. Guide B. Individual interviews for caregivers of children and adolescents previously engaged with armed groups. Guide C. Individual interviews for caregivers of children and adolescents previously engaged with armed groups.
(DOCX)

**S2 Appendix. Overview of key findings by country.** Table A. Key findings for adolescents by country: Risk and protective factors for engaging with armed groups and reintegrating with communities before, during, and after conflict. Table B. Key findings for caregivers by country: Risk and protective factors for adolescents engaging with armed groups and reintegrating with communities before, during, and after conflict.
(DOCX)

## Acknowledgments

The research team would like to acknowledge the International Rescue Committee Democratic Republic of Congo and Central African Republic teams who were instrumental in supporting the research design, data collection, and analysis, including: Ange Mashagiro, Elisabeth Sikulu, Julia Wendt Ulrike, Patricia Zawadi, Fabienne Zumbassa, Jerusha Julius Bode, Kennedy Atiya, Neema Alice, Justin Kambale, Ishara Josue, Esperance Twizere, and Nicodem Bizoza. We also extend sincere gratitude to the girls, boys, women, men, and community members who took part in the research and made this study possible.

## Author Contributions

**Conceptualization:** Alexandra H. Blackwell, Yvonne Agengo, Kathryn Falb.

**Data curation:** Alexandra H. Blackwell.

**Formal analysis:** Alexandra H. Blackwell, Daniel Ozoukou.

**Investigation:** Alexandra H. Blackwell, Kathryn Falb.

**Methodology:** Alexandra H. Blackwell, Kathryn Falb.

**Project administration:** Alexandra H. Blackwell, Yvonne Agengo, Julia Ulrike Wendt, Alice Nigane, Paradis Goana, Bertin Kanani.

**Supervision:** Kathryn Falb.

**Validation:** Julia Ulrike Wendt, Alice Nigane, Paradis Goana, Bertin Kanani.

**Visualization:** Alexandra H. Blackwell, Yvonne Agengo.

**Writing – original draft:** Alexandra H. Blackwell, Daniel Ozoukou.

**Writing – review & editing:** Yvonne Agengo, Daniel Ozoukou, Julia Ulrike Wendt, Alice Nigane, Paradis Goana, Bertin Kanani, Kathryn Falb.

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
