## [Decision Letter · Decision Letter 0]

24 Aug 2022

PGPH-D-22-01114

Drivers of ‘voluntary’ recruitment and challenges for families with adolescents engaged with armed groups: A conceptual framework from Central African Republic and Democratic Republic of the Congo

Dear Dr. Blackwell,

Thank you for submitting your manuscript to PLOS Global Public Health. After careful consideration, we feel that it has merit but does not fully meet PLOS Global Public Health’s publication criteria as it currently stands. Therefore, we invite you to submit a revised version of the manuscript that addresses the points raised during the review process.

EDITOR Comments: 

As you will see from the comments, the reviewers and I are impressed with the amount of work that has gone into this manuscript, and the sensitivity shown in presenting the experiences of caregivers and youth. Responding to the reviewers' questions and comments will strengthen the manuscript, and its potential for impact and informing practice.I agree with the reviewers' suggestions concerning clarifying the recruitment, data collection, and analysis methods, and would encourage the authors to complete the COREQ checklist to help with this. I also encourage the authors to include their data collection instruments and final codebook as supplementary materials. I also agree with the suggestions to revise the results to make the themes and sub-themes clearer to the reader, and that the conceptual figure models are a little hard to understand. Two of the reviewers provided their comments as an attachment, so please let the editorial staff know if you cannot view the attached comments.

We look forward to receiving your revised manuscript.

Kind regards,

Marie A. Brault, PhD

Academic Editor

Journal Requirements:

1. Please provide a/amend your detailed Financial Disclosure statement. This is published with the article. It must therefore be completed in full sentences and contain the exact wording you wish to be published.

2. Since your data is not available for proprietary reasons, please explain via email why the data is not available. Please also include the contact information for the third party organization that should be contacted should other researchers want to request access to this data and please include the full citation of where the data can be found. We also request that you verify with us via email that any researcher will be able to obtain the data set in the same manner that the you have obtained it. If you feel you are unwilling or unable to adhere to this policy, please explain your reasons by return email and your exemption request will be escalated to the editor for approval. Your exemption request will be handled independently and will not hold up the peer review process, but will need to be resolved should your manuscript be accepted for publication. One of the Editorial team will be in touch if they require more information.

Additional Editor Comments (if provided):

Please see above.

Reviewers' comments:

Reviewer's Responses to Questions

**Comments to the Author**

1. Does this manuscript meet PLOS Global Public Health’s publication criteria? Is the manuscript technically sound, and do the data support the conclusions? The manuscript must describe methodologically and ethically rigorous research with conclusions that are appropriately drawn based on the data presented.

Reviewer #1: Partly

Reviewer #2: Yes

Reviewer #3: Partly

2. Has the statistical analysis been performed appropriately and rigorously?

Reviewer #1: N/A

Reviewer #2: N/A

Reviewer #3: No

3. Have the authors made all data underlying the findings in their manuscript fully available (please refer to the Data Availability Statement at the start of the manuscript PDF file)?

Reviewer #1: No

Reviewer #2: No

Reviewer #3: Yes

4. Is the manuscript presented in an intelligible fashion and written in standard English?

Reviewer #1: Yes

Reviewer #2: Yes

Reviewer #3: No

5. Review Comments to the Author

Reviewer #1: First, thank you for your dedication to this population and the clear and obvious commitment displayed for their safety throughout the research process.

I agree with the authors' decision to not make the primary data publicly available based on concerns for safety and ethics. That said, in light of the journal's orientation towards replicability and transparency, it might be worth considering whether the qualitative instrumentation (interview guides) or analytical tools (code books, coding trees, etc) should be provided.

I would suggest pulling the contextual study setting information out before the methods section. Currently, this section feels to unnaturally break up the methodological information with historical and contextual information better suited for earlier in the manuscript.

I would also suggest providing all relevant information related to participant identification and recruitment in one place. Currently, there is rather vague recruitment information presented in "Study Participants". Additional information about recruitment and site selection were included in "Data Collection" and "Ethical and Safety Considerations".

Recruitment challenges were noted and changes to the process were mentioned. However, it is unclear what these changes entailed or how they improved recruitment.

I see that data were coded by one member of study team. Does this indicate that all interviews were coded by the same research team member or that each transcript was coded by only one person?

As presented, Tables 2 and 3 are difficult to follow. I am unsure if differences in word choice are reflect true nuance in the data, or if these are simply different phrases used to describe similar phenomena. For instance, is "no access to basic needs or livelihoods" distinct from "no access to income" and "caregivers unable to provide for needs".

Table 2 seems to suggest that in CAR a belief or affiliation with the group drives engagement, while the opposite seems to be true in the DRC (that a lack of that affiliation drives engagement). However, the reading of the results section seems to present a different story. Namely, that this affiliation just isn't a salient factor for engagement in the DRC.

I might suggest thinking through whether tables of the nature of Tables 2 and 3 are really the most appropriate way to present this qualitative data.

My biggest concern with this manuscript relates to the analytical process. Currently, the methodology reads as if themes were developed. But themes do not seem to be clearly presented. What appears under the heading of "theme" in Tables 2 and 3 appear to align better with the concept of theme categories related to the temporality lens the authors apply in the presentation of results. In the results section, themes are not clearly delineated, but the data seems to simply be described. Likewise, pieces of the analytical process seem to be borrowing from grounded theory. But without a theoretical framework for the analysis clearly stated, it is difficult to determine whether the analytical process described is appropriate.

Overall, the results seem to be getting at an important and potentially useful conceptual framework. However, the stories of the different groups get lost in the sheer volume of data being presented and the lack of organization to guide that presentation. There are both gendered and country-specific lenses being applied to the data. But the results often jump from those directly engaged to those at-risk of being engaged to caregivers and often back to adolescents of some kind. It's challenging to follow any one thread or narrative in its current form. I would suggest a more structured presentation utilizing clear headers and subheaders. For instance, you could discuss cross-cutting ideas (spanning both CAR and DRC), then move to discussing ideas that were contextually specific or distinct. Given that the paper's Figure 1 depends on data from all stakeholder groups, it likely does not make sense to break this paper out into more focused publications. However, I do recommend considering an organization approach that makes this volume of data more reader-friendly.

Reviewer #2: All my comments are included in the attachment. No other comments are written here.

Word Limit ------------------------------------------------------------------------------------------------------------

Reviewer #3: The authors explore well debated phenomenon of child and adolescent involved in armed conflicts, particularly the increasing self-enrolment or “voluntarily enrolment” which is contested by several scholars who rethink the concept of child soldiers from the perspectives of African rites of passage (initiation) to adulthood, which starts from twelve to full adulthood (Honwana 2006; Van Rooyen et al. 2006). The authors further explore the prevention of child and adolescents’ recruitment, their use as soldiers, re-remobilisation, their reintegration from the perspectives of their caregivers, and the extent to which family can lessen their chance of re-recruitment. The findings reveal a nexus of factors that maintain the vulnerability of boys and girls to recruitment and re-recruitment. Those factors include:

• Traumatic experiences and economic hardships keep children and adolescents vulnerable.

• Erosion of family protective family relationships endangers meaningful reintegration.

• Familial support would prevent re-recruitment.

• A better understanding of children’s experiences in armed conflict can incentivise resiliency of child soldiers and their support by caregivers.

• more comprehensive programming models can be developed to prevent voluntary , etc.

Lastly,

• the authors highlight the necessity of a family-based approach to ameliorate adolescent resilience to recruitment by armed groups.

• They advocate additional programs of caregivers’ support in armed conflict settings to increase their own mental health and family functionality.

• Better support structures are also called for to prevent recruitment, facilitate a successful reintegration of returnee young soldiers, and minimise the danger of re-mobilisation, tec.

While the manuscript shows a lot of potential, especially the finding if brings forth, the paper has serious weaknesses which include:

• Lack of clear research design to outline how the research will be conduct and who will be participating and what they will be contributing to generate knowledge.

• The conceptual framework is not well defined and explained as far as voluntary recruitment of different age groups are concerned.

• There is little engagement with existing legal frameworks and literature pertaining to child and adolescent recruitment.

• The lack of a well explained theoretical framework makes it hard to follow the orientation of the inquiry.

Additional comments and advice are subsequently provided to ameliorate the scientific value of this paper.

The paper requires major corrections.

6. PLOS authors have the option to publish the peer review history of their article (what does this mean?). If published, this will include your full peer review and any attached files.

**Do you want your identity to be public for this peer review?** For information about this choice, including consent withdrawal, please see our Privacy Policy.

Reviewer #1: No

Reviewer #2: No

Reviewer #3: No

---

## [Decision Letter · Decision Letter 1]

8 Feb 2023

PGPH-D-22-01114R1

Drivers of ‘voluntary’ recruitment and challenges for families with adolescents engaged with armed groups: A conceptual framework from Central African Republic and Democratic Republic of the Congo

Dear Dr. Blackwell,

Thank you for submitting your manuscript to PLOS Global Public Health. After careful consideration, we feel that it has merit but does not fully meet PLOS Global Public Health’s publication criteria as it currently stands. Therefore, we invite you to submit a revised version of the manuscript that addresses the points raised during the review process.

Editor Comments:

We appreciate the authors' careful attention to the comments. The manuscript is much improved, and the requested revisions are relatively minor clarifications.Specifically, please address the comments related to the qualitative analysis, and clarify how grounded theory was employed. For Reviewer 4's comments concerning how to describe the locations and programs involved in the project, I defer to the authors' own IRB and advisory group requirements, given the sensitivity of the data and the need to protect participants and communities. I do agree that the "[REDACTED]" reads a little awkwardly, and simply referring to generic child protection programs (or another general description) would be sufficient. I also think that the definitions regarding adolescents and young adults are clearer, but check the manuscript to ensure the terminology is consistent throughout.

We look forward to receiving your revised manuscript.

Kind regards,

Marie A. Brault, PhD

Academic Editor

Journal Requirements:

3. We have noticed that you have a list of Supporting Information legends in your manuscript. However, there are no corresponding files uploaded to the submission. Please upload them as separate files with the item type 'Supporting Information'. 

Additional Editor Comments (if provided):

See above.

Reviewers' comments:

Reviewer's Responses to Questions

**Comments to the Author**

1. If the authors have adequately addressed your comments raised in a previous round of review and you feel that this manuscript is now acceptable for publication, you may indicate that here to bypass the “Comments to the Author” section, enter your conflict of interest statement in the “Confidential to Editor” section, and submit your "Accept" recommendation.

Reviewer #1: (No Response)

Reviewer #4: (No Response)

2. Does this manuscript meet PLOS Global Public Health’s publication criteria? Is the manuscript technically sound, and do the data support the conclusions? The manuscript must describe methodologically and ethically rigorous research with conclusions that are appropriately drawn based on the data presented.

Reviewer #1: Yes

Reviewer #4: (No Response)

3. Has the statistical analysis been performed appropriately and rigorously?

Reviewer #1: N/A

Reviewer #4: N/A

4. Have the authors made all data underlying the findings in their manuscript fully available (please refer to the Data Availability Statement at the start of the manuscript PDF file)?

Reviewer #1: Yes

Reviewer #4: (No Response)

5. Is the manuscript presented in an intelligible fashion and written in standard English?

Reviewer #1: Yes

Reviewer #4: Yes

6. Review Comments to the Author

Reviewer #1: Publication requirements are met and detailed explanations regarding data availability have been provided by the authors.

First, I appreciate the authors' diligence in responding to the first round of reviewer comments. The responses were thoughtful and collegial, and the edits improved clarity of the paper.

I still have some concerns with the ways that the qualitative paradigm, methodology, and analytical methods are described. After reading the manuscript, appendices, and point-by-point comments, I feel as if the methodology is likely sound but is not being communicated in a way that shows the paradigmatic and methodological underpinnings. The use of in-depth interviews and the narrative lifeline elicitation technique is clear. However, narrative inquiry (a methodology) is mentioned in the analysis section. It is noted that the methodology guided "data collection and analysis". As a result, it should be placed earlier in the methods section so that the influence of narrative inquiry on the methods and data collection tools is clearer.

A second concern is related to the brief mention of grounded theory related to coding. Grounded theory encompasses much more than just the use of inductive coding, though that appears to be what is described by the authors here. I recommend interrogating the use of grounded theory or simply developed inductive codes. In its current form, I do not see the other elements of GT in the methodological framework. If such usual hallmarks were present in the study design, I recommend making these clearer. If they were not, it may be more accurate to state that inductive codes were used, without suggesting the use of GT.

Consistent with the SRQR reporting guidelines, consider a title that communicates the qualitative nature of the data.

Reviewer #4: Thank you for studying this important subject and carefully conduct research with such a vulnerable population group.

Here a few general comments and then others below by sections

1. Whether to present the findings as a (new?) framework

- I work in public health in conflict affected settings, but I am not familiar with the literature around voluntary child recruitment; it is therefore unclear to me if the authors use an existing framework or present a new one (which would justify the title). The four levels seem to be drawn and adapted from existing socio ecological frameworks. If the time component is the novelty, I would suggest to further describe and justify it. As it is now, it is mentioned once (line 306) but the rational why it should consider is not explained, i.e. why do you argue that factors evolve over time of the conflict?

- It is therefore necessary to clarify whether you defined a new framework & what the novelty is, or whether you frame your results adapting/ using framework XY.

- Regarding the time component, given that the insecurity situation continues in the two sites, I find it confusing to call the third phase “Post conflict”. I wonder whether these should be rather defined as “pre-recruitment”, “during recruitment”, “after involvement” (as you did in the pre-print version of the paper available online).

2. The issue of using “children” or “adolescents”

- Reviewer 3 (page 106) had previous raised the concern about the use of the words “Children” and “adolescents” (almost) interchangeably. There are still instances where only one of the two is used, and it is unclear if the other group is not included. It may be less confusing to use only one, or to clearly define the two groups at the beginning and then use each word appropriately. I would suggest in any case to add the definition of your study population from the introduction. As there is no legal definition of adolescents, you may want to stick to children. Or as respondents were finally aged 14 and above you may decide to use only adolescents.

o For example, page 7, first sentence of Methods: “In depth qualitative interviews were conducted with adolescents and caregivers” (so no children?)

- Also, it is confusing why respondents 18 to 20 were included, given that they are no longer (legally) children. You do state that they were asked about their experience when recruited as adolescents under the age of 18, but it may be good to know how many of the respondents belonged to each age group.

3. Ensuring confidentiality and protection of the study population.

While I recognize the importance of this, using the word “REDACTED” is confusing (when reading) and I think not sufficient:

- As the majority of the authors work for IRC, the reader will likely quickly assume that IRC is implementing a child protection program and that participants come from this program. Looking up the “Safe Healing and Learning Spaces” (that you mention on page 11) leads you to the IRC website.

- I would suggest to state that neither precise locations nor names of the child protection organizations working in each country will be made for security and confidentiality reasons. You can then remove the “redacted” and just refer to “the child protection organization”.

- I would suggest to keep the locations as Ouham Pende (as it is a prefecture, I assume it is big enough), and Nord Kivu (without stating Nyriagongo like on page 11).

- From a methodological point of view and for the sake of transparency I think it is important to state that IRC implements child protection projects in the areas to exclude (or not) possible recruitment biases.

- I leave it to the editor to decide how to best deal with this.

Methods

- Inclusion criteria:

o Line 174: only “formerly” is stated while Line 176 “formerly or currently involved” � please confirm if it is only formerly or also currently.

- Line 177: Languages spoken: Swahili and Kinyarwanda are mentioned, which I assume apply to DRC. Need to add Sango for CAR to align with line 253.

- Sample size:

o Could you clarify if the calculations were made for the children (only) or for both children and caregivers?

o It sounds as though you had to modify the sample size calculation (i.e., I assume reducing the sample size?) from your initial approach as you had challenges in recruiting girls and at risk children.

o The second (revised approach) proposes to sample 10% of the verified cases. You state there are

442 cases in DRC � target sample size 45

75 cases in CAR � however the target sample size is 35 and is based on “310 cases of grave violations”. It seems the word case is used for children in DRC and for events in CAR? could you provide a definition of a case?

o How did you decide to interview 39 caregivers?

- Table 1: “out of armed groups” does it mean that none of the respondents were currently involved?

- Ethical consideration:

o Suggest moving lines 160 to 171 to this section.

- Coding:

o Line 314 (manuscript) vs response to reviewer 1 (page 92): it seems misleading to state that data were coded by two members, when you answered that it was coded by one member except for the initial 5% used to ensure coordination of coding. I would therefore say that “two members coded 5% of the transcripts to harmonize approaches, after which one team member coded all the remaining transcripts”.

Results

- Line 349: “also” � this is the first factor that is reported, better to remove “also”.

- Line 358: remove CAR, as data were collected in March 2020 in CAR, ie. very early in the COVID pandemic.

- To be formatted as quotes:

o Lines 385 to 387

o Lines 479 - 481

o Lines 518 - 520

o Lines 524 – 526

o Lines 558 – 560

o Lines 576 – 577

o Lines 609 – 611

o Lines 612 - 613

o Lines 697 - 699

- Lines 497-501: the quote does not support the statement made on lines 497, rather says the opposite.

Discussion

- Fig 1: you should add in the methodology how you defined and counted the occurrence of certain factors, i.e. “the numeric findings” behind this figure, and how it is reflected in the size of the circles. Is this something that was done in the software you used?

- The discussion is long and includes some repetitions from the results. As the result section is long, I would suggest cutting down the discussion and keep only key points.

7. PLOS authors have the option to publish the peer review history of their article (what does this mean?). If published, this will include your full peer review and any attached files.

**Do you want your identity to be public for this peer review?** For information about this choice, including consent withdrawal, please see our Privacy Policy.

Reviewer #1: No

Reviewer #4: No

---

## [Decision Letter · Decision Letter 2]

12 Apr 2023

Drivers of ‘voluntary’ recruitment and challenges for families with adolescents engaged with armed groups: Qualitative insights from Central African Republic and Democratic Republic of the Congo

PGPH-D-22-01114R2

Dear Miss Blackwell,

We are pleased to inform you that your manuscript 'Drivers of ‘voluntary’ recruitment and challenges for families with adolescents engaged with armed groups: Qualitative insights from Central African Republic and Democratic Republic of the Congo' has been provisionally accepted for publication in PLOS Global Public Health.

Best regards,

Marie A. Brault, PhD

Academic Editor

Reviewer Comments (if any, and for reference):

Reviewer's Responses to Questions

**Comments to the Author**

1. If the authors have adequately addressed your comments raised in a previous round of review and you feel that this manuscript is now acceptable for publication, you may indicate that here to bypass the “Comments to the Author” section, enter your conflict of interest statement in the “Confidential to Editor” section, and submit your "Accept" recommendation.

Reviewer #1: All comments have been addressed

Reviewer #4: (No Response)

2. Does this manuscript meet PLOS Global Public Health’s publication criteria? Is the manuscript technically sound, and do the data support the conclusions? The manuscript must describe methodologically and ethically rigorous research with conclusions that are appropriately drawn based on the data presented.

Reviewer #1: Yes

Reviewer #4: Yes

3. Has the statistical analysis been performed appropriately and rigorously?

Reviewer #1: N/A

Reviewer #4: N/A

4. Have the authors made all data underlying the findings in their manuscript fully available (please refer to the Data Availability Statement at the start of the manuscript PDF file)?

Reviewer #1: No

Reviewer #4: (No Response)

5. Is the manuscript presented in an intelligible fashion and written in standard English?

Reviewer #1: Yes

Reviewer #4: Yes

6. Review Comments to the Author

Reviewer #1: All comments have been adequately addressed, and the manuscript appears to be considerably strengthened over the prior two submissions.

Reviewer #4: Thank you for revising the paper and addressing my comments.

Three minor and final comments:

Line 145: can you spell out CAAFAG as it is the first time you use the acronym?

Line 227: does it mean that you oversample adolescent girls in DRC to compensate for difficulties in recruiting adolescent girls in CAR? Clarifying this would make the sentence easier to understand.

Line 229: Can you explain what you mean with “based on the caseload for each country”? as the caseload in each country is different, readers may expect a different number of caregivers, but then you say 20 in each country. Given that it is a qualitative study, you could say that you aimed for 50% of the caregivers?

7. PLOS authors have the option to publish the peer review history of their article (what does this mean?). If published, this will include your full peer review and any attached files.

**Do you want your identity to be public for this peer review?** For information about this choice, including consent withdrawal, please see our Privacy Policy.

Reviewer #1: **Yes: **Hannah L. N. Stewart

Reviewer #4: No
